# Mechanisms of substrate recognition by a typhoid toxin secretion-associated muramidase

**Tobias Geiger[1†], Maria Lara-Tejero[1†], Yong Xiong[2], Jorge E Galán[1]\***

[1]Department of Microbial Pathogenesis, Yale University School of Medicine, New Haven, United States; [2]Department of Molecular Biophysics and Biochemistry, Yale University School of Medicine, New Haven, United States

**Abstract** Typhoid toxin is a virulence factor for the bacterial pathogen *Salmonella* Typhi, which causes typhoid fever in humans. After its synthesis by intracellular bacteria, typhoid toxin is secreted into the lumen of the *Salmonella*-containing vacuole by a secretion mechanism strictly dependent on TtsA, a specific muramidase that facilitates toxin transport through the peptidoglycan layer. Here we show that substrate recognition by TtsA depends on a discrete domain within its carboxy terminus, which targets the enzyme to the bacterial poles to recognize YcbB-edited peptidoglycan. Comparison of the atomic structures of TtsA bound to its substrate and that of a close homolog with different specificity identified specific determinants involved in substrate recognition. Combined with structure-guided mutagenesis and in vitro and in vivo crosslinking experiments, this study provides an unprecedented view of the mechanisms by which a muramidase recognizes its peptidoglycan substrate to facilitate protein secretion.

**\*For correspondence:**
jorge.galan@yale.edu

[†]These authors contributed equally to this work

**Competing interests:** The authors declare that no competing interests exist.

## Introduction

Most virulence factors must be secreted from bacterial cells so that they can interact with host determinants and aid bacterial replication within the host. Consequently, bacterial pathogens have evolved a myriad of protein secretion mechanisms specifically tailored for the virulence factors they secrete (*Christie, 2019*; *Galán and Waksman, 2018*; *Costa et al., 2015*). Typhoid toxin is an important virulence factor for the human pathogen *Salmonella* Typhi (*Galán, 2016*), the cause of typhoid fever in humans and a major public health concern (*Parry et al., 2002*; *Crump and Mintz, 2010*; *Raffatellu et al., 2008*; *Wain et al., 2015*; *Dougan and Baker, 2014*). This toxin is unique in that is only produced by intracellular bacteria (*Haghjoo and Galán, 2004*; *Fowler and Galán, 2018*). After its synthesis, the toxin is secreted into the lumen of the *Salmonella*-containing vacuole, packaged into vesicle carriers, and subsequently transported to the extracellular environment from where it can reach its target cells (*Spanò et al., 2008*; *Chang et al., 2016*; *Chang et al., 2019*). Typhoid toxin secretion from the bacterial cell is mediated by a unique transport mechanism that requires the activity of TtsA, a specialized muramidase that facilitates the translocation of typhoid toxin from the *cis* to the *trans* side of the peptidoglycan (PG) layer of the bacterial envelope (*Hodak and Galán, 2013*; *Geiger et al., 2018*). The bacterial PG is composed of glycan strands made up of N-acetylglucosamine (GlcNac) and N-acetylmuramic acid (MurNac) linked by β-(1, 4) glycosidic bonds (*Turner et al., 2014*; *Egan et al., 2015*; *Vollmer et al., 2008*). These strands are connected by short peptides composed of L- and D-amino acids, which are cross-linked to one another by specific transpeptidases. Most of these cross-links are between the carboxyl group of D-Ala at position 4 of one peptide to the ε-amino group of the m-Dap residue at position 3 of another (D-D cross-links) (*Glauner et al., 1988*). However, although less abundant, crosslinks can also occur between two m-Dap residues of adjacent stem peptides (L-D cross-links) (*Glauner et al., 1988*; *Höltje, 1998*;

*Quintela et al., 1997*). In previous studies we have shown that TtsA is unique in that it exerts its function at the bacterial poles, and it requires the activity of YcbB, a bacterial transpeptidase that is responsible for introducing L-D crosslinks to the peptides that link the glycan strands of the peptidoglycan layer (*Geiger et al., 2018*). As the L-D-cross-links are most likely limited to specific subdomains of the PG layer, the substrate specificity exhibited by TtsA is thought to topologically restrict its activity. Consistent with this hypothesis, TtsA-mediated PG remodeling and typhoid toxin secretion occurs exclusively at the bacterial poles (*Geiger et al., 2018*). Although essential for the understanding of its function, there is no information about the mechanisms by which TtsA recognizes its substrate and localizes to the bacterial poles. In this study we have defined the specific domain of TtsA that targets its activity to the L-D-cross-linked PG at the bacterial poles. We have also solved the atomic structure of TtsA bound to its substrate and that of a close homolog with different specificity, which has allowed us to define the structural bases for its substrate specificity. Overall, these studies provide an unprecedented view of the mechanisms by which a bacterial muramidase involved in protein secretion engages its substrate.

## Results

### The carboxy-terminal but not the catalytic domain of TtsA confers substrate specificity and typhoid toxin secretion functions

We have previously shown that Sen1395, a close TtsA homolog encoded by *Salmonella* Enteritidis (*Figure 1a*), was unable to complement a *S.* Typhi Δ*ttsA* mutant strain for its ability to form typhoid-toxin vesicle carriers in infected cells (*Hodak and Galán, 2013*), an indirect measure of typhoid toxin in vivo secretion from the bacterial cell (*Spanò et al., 2008*). Consistent with this observation we found that, when we replaced *ttsA* in the *S.* Typhi chromosome with *sen1395*, the resulting strain was unable to mediate the translocation of typhoid toxin from the *cis* to the *trans* side of the PG layer (*Figure 1b*). To obtain insight into the potential mechanisms by which TtsA recognizes its substrate we constructed a series of chimeric proteins between TtsA and Sen1395, all expressed at equivalent levels to those of wild type (*Figure 1—figure supplement 1*). We then examined the ability of the different constructs to complement a *S.* Typhi Δ*ttsA* mutant for its ability to translocate typhoid toxin through the PG layer. We found that a chimera made up of the amino terminal half of Sen1395 (amino acids 1–92), which contains its catalytic domain, and the carboxy-terminal half of TtsA (amino acids 93–180) was able to fully complement Δ*ttsA S.* Typhi mutant strain for the translocation of typhoid toxin across the PG layer (*Figure 1b* and *Figure 1—figure supplement 2*). These results are consistent with our previous studies indicating that a similar construct restored the formation of vesicle carrier intermediates in cells infected with a Δ*ttsA S.* Typhi mutant strain (*Hodak and Galán, 2013*). Construction of an additional chimera with a larger proportion of Sen1395 resulted in a reduction in its ability to complement the *S.* Typhi Δ*ttsA* mutant for typhoid toxin translocation across the PG layer (*Figure 1b*). These findings support the notion that the substrate-binding and protein secretion specific functions of TtsA are contained within its carboxy terminal domain. More broadly, the observation that Sen1395 cannot complement a Δ*ttsA S.* Typhi mutant for typhoid toxin secretion but that its catalytic domain can be exchanged with TtsA's without affecting protein secretion function suggests that differences in substrate-recognition rather than substrate catalysis are responsible for Sen1395's inability to mediate typhoid toxin secretion.

We have previously shown that TtsA's muramidase activity exhibits exquisite specificity for L-D cross-linked PG (*Geiger et al., 2018*). TtsA is therefore unable to hydrolyze PG obtained from a *S.* Typhi mutant lacking *ycbB*, which encodes the only transpeptidase in this bacterium capable of introducing L-D cross-links to the PG structure (*Magnet et al., 2008*). Consequently, over-expression of full-length TtsA in wild-type *S.* Typhi halted its growth (due to PG hydrolysis), but its overexpression in the Δ*ycbB S.* Typhi mutant did not (*Figure 1c* and *Figure 1—figure supplement 3*). Consistent with the hypothesis that the carboxy-terminus of TtsA confers its substrate specificity, over expression of a chimeric construct consisting of the amino-terminal catalytic domain (1-92) of Sen1395 and the carboxy-terminal substrate binding domain (93-180) of TtsA was able to halt the growth of wild-type *S.* Typhi but not the growth of the Δ*ycbB* mutant strain (*Figure 1c*) despite equivalent levels of expression (*Figure 1—figure supplement 3*). In contrast, overexpression of full-length Sen1395 was able to halt the growth of both wild type and the Δ*ycbB* mutant strains (*Figure 1c*). We have

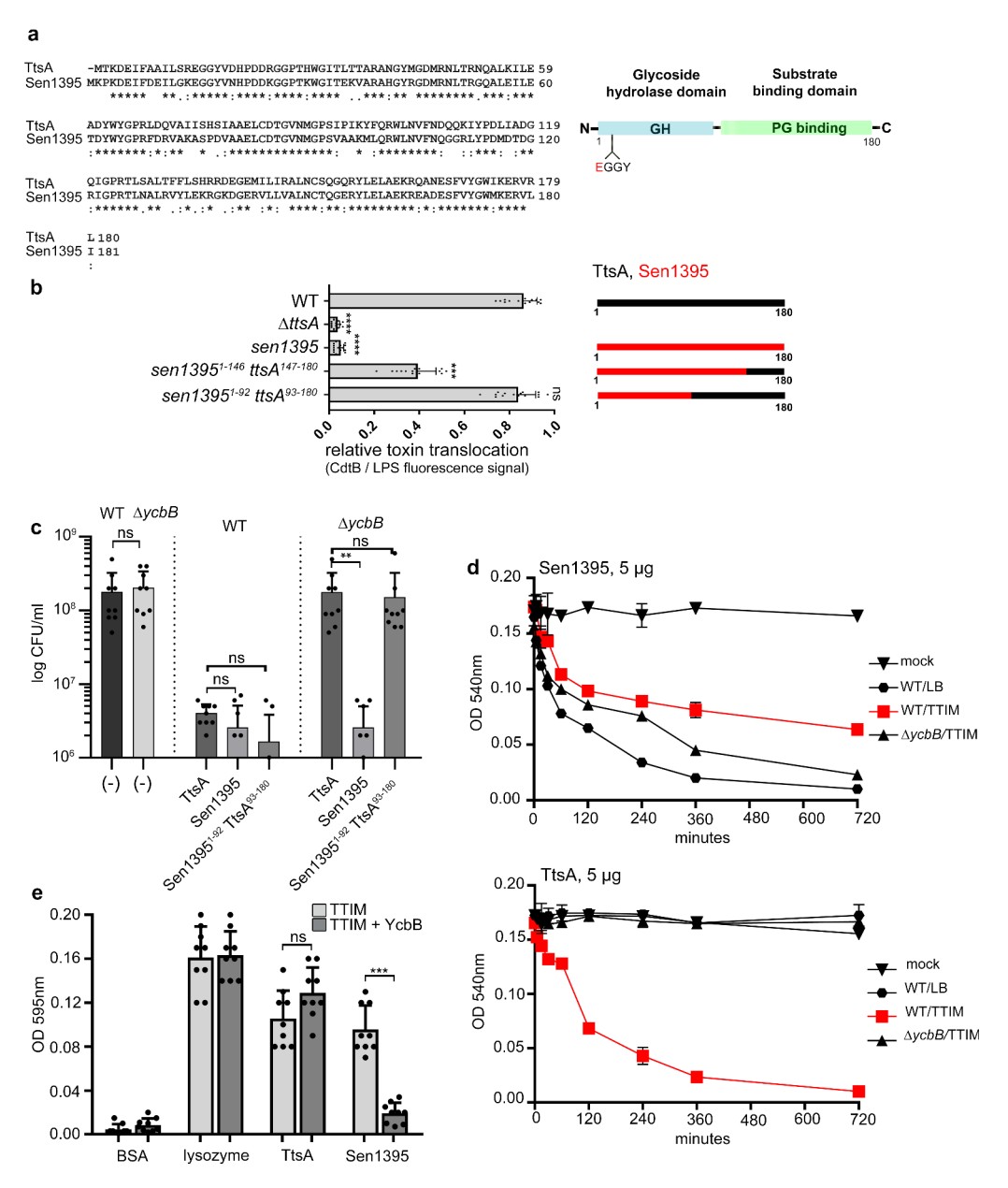

**Figure 1.** The carboxy-terminal domain of TtsA confers substrate specificity and typhoid toxin secretion functions. (a) Amino acid sequence alignment between TtsA and its *S.* Enteritidis homolog Sen1395. Identical (*) and conserved (:) residues are indicated. The predicted domain organization is also shown. (b) Ability of *S.* Enteritidis Sen1395 and the indicated chimeras to complement a *S.* Typhi Δ*ttsA* mutant strain for typhoid toxin translocation across the PG. The relative amount of typhoid toxin translocation across the PG was quantified by immunofluorescence microscopy after staining with antibodies to the FLAG epitope (to visualize CdtB, a component of typhoid toxin), and LPS (to visualize bacterial cells). The average ratios of typhoid toxin- and LPS-associated fluorescence intensity ± standard deviation are shown (****p<0.0001, ***p<0.001, ns p=0.395, two-sided Student's *t* Test) (*Figure 1—source data 1*). For each experiment a total of 10 images were collected from which 100 randomly selected bacteria per image were analyzed. A diagram of the different constructs is also shown. (c) Effect of expression of TtsA, Sen1395 or a TtsA-Sen1395 chimera on the growth of wild type or Δ*ycbB S.* Typhi, as indicated. Overnight grown bacteria were subcultured (1:50) in TTIM and grown to an $OD_{600}$ of 0.3, at which point 0.3% arabinose was added to the bacterial cultures to induce the expression of the different muramidases, and subsequently incubated for 20 hr at 37°C. Colony forming units (CFUs) were determined by plating bacterial dilutions on LB agar plates. Values represent the mean + /- standard deviation (**p<0.01, ns

*Figure 1 continued on next page*

*Figure 1 continued*
p=0.6892, p=0.0115, p=0.0145, p=0.7505, two-sided Student's *t*-Test, when compared to the values of the respective control strains, shown on the left bars). (**d-e**) TtsA and Sen1395 muramidase activity on YcbB-edited PG. Peptidoglycan was isolated from wild-type *S.* Typhi grown in either LB, or TTIM, or from the Δ*ycbB* *S.* Typhi mutant grown in TTIM, as indicated (**d**). The PG hydrolytic activity of purified Sen1395 and TtsA was evaluated using a turbidimetric assay. Graphs show the mean turbidity (measured at OD$_{540}$ nm) ± standard deviation (**d**). Alternatively, PG was isolated from wild-type *S.* Typhi or from a wild-type *S.* Typhi strain carrying a plasmid over-expressing *ycbB* under the control of an arabinose-inducible promoter (**e**). Both strains were grown for 24 hr in TTIM containing 0.3% arabinose. The PG hydrolytic activity of purified Sen1395, TtsA, lysozyme (positive control) and BSA (negative control) was evaluated using a Remazol Brilliant Blue (RBB)-dye release assay. The dye bound to PG, is released to the supernatant due to PG hydrolysis and can be measured by its absorbance. Graphs show the mean absorbance (measured at OD$_{595}$ nm)± standard deviation (\*\*\*p<0.0026, ns p=0.6892, two-sided Student's t-Test). (**b–e**) All data are derived from at least three independent experiments (*Figure 1—source data 1*).
The online version of this article includes the following source data and figure supplement(s) for figure 1:

**Source data 1.** Contains source data related to *Figure 1b–e*.
**Figure supplement 1.** Western blot analysis of the expression levels of CdtB and TtsA in the indicated *S.*
**Figure supplement 2.** Ability of *S. Enteritidis* Sen1395 and the indicated chimera to complement a *S.* Typhi Δ*ttsA* mutant strain for typhoid toxin translocation across the PG.
**Figure supplement 3.** Western blot analysis of the expression levels of the indicated plasmid-born FLAG-tagged TtsA, Sen1395 or chimeric proteins expressed from an arabinose-inducible promoter in the *S.* Typhi strains assayed in *Figure 1C*.

previously shown that the proportion of LD crosslinks increases significantly when bacteria are grown in TTIM (*Geiger et al., 2018*), a culture medium that mimics conditions found by intracellular *Salmonella*. Consequently, while purified TtsA is not able to hydrolyze PG isolated from LB grown bacteria, it is able to effectively hydrolyze PG obtained after S. Typhi growth in TTIM (*Geiger et al., 2018*) (*Figure 1d*). We found that purified Sen1395 was able to hydrolyze PG isolated from both wild type and Δ*ycbB* *S.* Typhi, after growth in LB or TTIM, although the latter to a lesser extent (*Figure 1d*). However, purified Sen1395 was unable to hydrolyze PG isolated from a strain overproducing YcbB and thus containing an increased proportion of LD cross-links (*Figure 1e*). Taken together, these results indicate that Sen1395 and TtsA exhibit different PG specificity and that the substrate specificity determinants of both TtsA and Sen1395 are contained within their carboxy-terminal half.

## TtsA polar localization and substrate recognition determinants are distinct

We have previously shown that TtsA localizes to and exerts its activity at the bacterial poles (*Geiger et al., 2018*). To gain insight into the mechanisms that target TtsA to the bacterial poles, we first investigated the subcellular localization of Sen1395, which, as shown above, has different substrate specificity and is unable to complement a *S.* Typhi Δ*ttsA* mutant for typhoid toxin secretion. We found that, unlike TtsA, when expressed in *S.* Typhi Sen1395 was not enriched at the poles and showed a uniformed distribution around the bacterial surface (*Figure 2a and b* and *Figure 2— figure supplement 1*). We then tested the subcellular localization of different TtsA-Sen1395 chimeric constructs. We found that the TtsA carboxy-terminus is essential for polar localization since a chimera consisting of the first 92 amino acids of Sen1395 and amino acids 93 to 180 of TtsA localized to the pole (*Figure 2c*). Furthermore, we found a strict correlation between the ability of these constructs to complement a *S.* Typhi Δ*ttsA* mutant for typhoid toxin PG translocation (*Figure 1b*) and their polar localization (*Figure 2a and c*), supporting the notion that polar localization is essential for TtsA protein secretion function. We also examined the ability of the different chimeras to mediate PG remodeling at the bacterial poles. We labeled the *S.* Typhi PG using an alkyne-modified D-alanine and an azide-containing fluorophore that can be linked by click chemistry. The short peptides that link the PG glycan strands contain D-alanine residues. Therefore, the metabolic incorporation of alkyne-modified D-alanine and its subsequent linking to an azide-containing fluorophore can be used to label remodeling PG (*Siegrist et al., 2013*; *Cameron et al., 2014*; *Kuru et al., 2015*). We found that expression of the chimeric constructs containing the carboxy-terminus of TtsA and therefore targeted to the bacterial poles, strictly correlated with the presence of PG remodeling at these

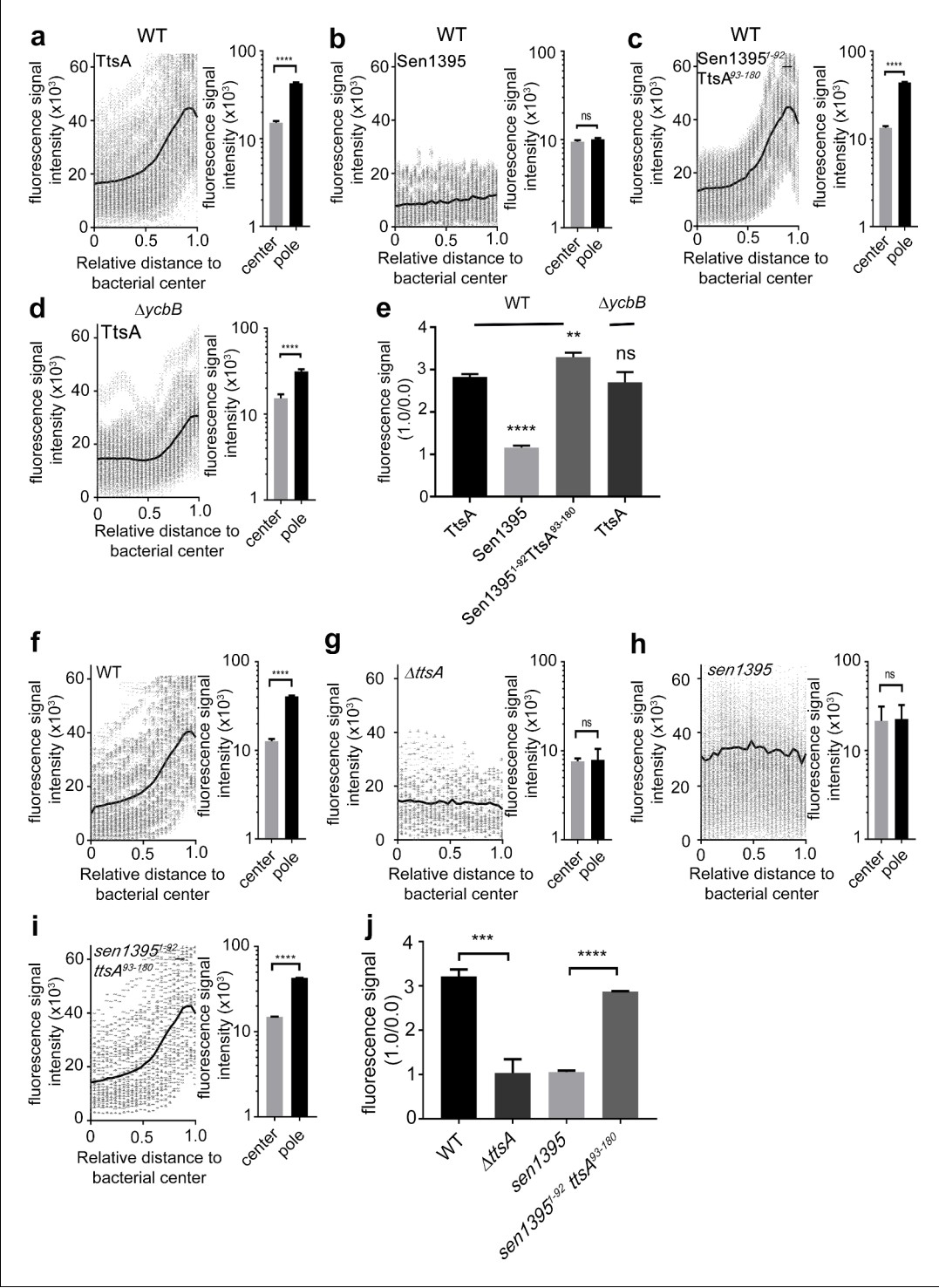

**Figure 2.** The carboxy-terminus of TtsA controls polar localization. (**a–e**) Subcellular distribution of TtsA, Sen1395, and chimeric proteins. Wild type (**a–c**) or Δ*ycbB* (**d**) *S*. Typhi strains carrying chromosomally-encoded FLAG-tagged TtsA (**a and d**), Sen1395 (**b**) or the chimeric Sen1395-TtsA protein (**c**) were grown for 24 hr in TTIM, fixed, and stained with a mouse antibody directed to the 3xFLAG-epitope (green) (to visualize the muramidases) and a rabbit antibody directed to *S*. Typhi LPS (red). The scatter plots show fluorescence signal intensities for the indicated FLAG-tagged protein distributed along the axes of individual bacterial cells. Twenty-six measuring points were defined from the center (0) to each of the poles (1.0) of the bacterial cells and the distribution of the fluorescence intensity along the axes of bacteria were analyzed with the MicrobeJ plug-in of ImageJ (https://imagej.nih.gov/ij).
*Figure 2 continued on next page*

*Figure 2 continued*

The black line depicts the average of the intensities measured at each of the 26 measuring points. Data in each panel are from 1800 individual measurements at each of the measuring points from 900 bacteria analyzed in opposite directions from the center. The bar graphs next to each scatter plot show the quantification of the signal intensities at the measuring point furthest from the center (1.0), and at the center of each bacterium (0). Data represent the mean ± standard deviation from 1800 measurements (****p<0.0001, ns p=0.0864, two-sided Student's t-Test) (a–d). (e) Bar graph shows the average ratios between the signal intensity of the indicated proteins measured at the furthest point from the center (1.0) and at the center of each bacterium (0.0) Data represent the mean ± standard deviation (****p<0.0001, **p<0.01, ns p=0.4369, two-sided Student's *t*-Test). (f–j) TtsA- and Sen1395-mediated PG remodeling. *S.* Typhi wild type expressing *ttsA* (f), its isogenic Δ*ttsA* mutant (g), or *S.* Typhi strains expressing either *sen1395* (h) or a *sen1395-ttsA* chimera (i) were grown in TTIM for 24 hr and the PG was metabolically labeled with alkyne-D-alanine. Remodeling PG was subsequently revealed with azido-AF488 after its linkage to the alkyne-D-alanine that had been incorporated into the PG layer. The scatter plots show the results of line scan analyses of fluorescence signal intensities along the axes of individual bacterial cells as described above. The line depicts the average fluorescence for each measured point. The bar graphs next to each scatter plot show the average ratios of the signal intensities measured at the furthest point from the center (1.0) and at the center of each bacterium (0.0). Data represent the mean ± standard deviation from 1800 measurements (****p<0.0001, ns p=0.8464 and p=0.8993, two-sided Student's t-Test) (f–i). (j) Bar graph shows the average ratios between the signal intensity measured at the point furthest from the center (1.0) and at the center of each bacterium (0.0). Data represent the mean ± standard deviation (***p<0.001, ****p<0.0001, two-sided Student's t-Test). (a–j) All data are derived from at least three independent experiments (*Figure 2—source data 1*). The online version of this article includes the following source data and figure supplement(s) for figure 2:

**Source data 1.** Contains source data related to *Figure 2a and c*.
**Figure supplement 1.** Subcellular distribution of TtsA, Sen1395, and the indicated Sen1395-TtsA chimera.
**Figure supplement 2.** TtsA- and Sen1395-mediated PG remodeling.

---

sites (*Figure 2c and d* and *Figure 2—figure supplement 2*). Since this domain is also associated with TtsA's ability to hydrolyze L-D-cross-linked PG, we specifically tested the possibility that this localized PG modification at the bacterial poles introduced by the polarly-localized YcbB transpeptidase may serve as polar-localization signal. We reasoned that if this was the case, TtsA should not be able to localize to the poles in a *S.* Typhi Δ*ycbB* mutant strain, which lacks L-D-cross-linked PG (*Geiger et al., 2018*). However, we found that the polar localization of TtsA in the Δ*ycbB* mutant was indistinguishable from wild type (*Figure 2a and d*) indicating that the presence of L-D-cross-linked PG per se is not essential for the polar localization of TtsA.

We have previously shown that changing asparagine at position 166 of TtsA to aspartic acid (N166D), which is the residue present in Sen1395 at the equivalent position, impaired the formation of typhoid toxin vesicle carrier intermediates in *S.* Typhi infected cells, an indirect measurement of toxin secretion (*Hodak and Galán, 2013*). Consistent with these findings we found that this mutation severely impeded TtsA-mediated typhoid toxin translocation across the PG layer (*Figure 3a* and *Figure 3—figure supplement 1*). Conversely, introduction of an asparagine at the same position in Sen1395 (D166N) conferred on this protein homolog the ability to partially complement typhoid toxin translocation across the PG in a Δ*ttsA S.* Typhi mutant strain (*Figure 3a* and *Figure 3—figure supplement 1*). Taken together, these observations indicate that TtsA N166 plays an important role in conferring on this muramidase the ability to translocate typhoid toxin across the PG layer. Since the typhoid toxin translocating activity of TtsA strictly correlates with its ability to localize to the bacterial poles, we tested the subcellular localization of TtsA$^{N166D}$. We found that this mutant localized to the bacterial poles in a manner indistinguishable from wild type (*Figure 3b and c*) suggesting that N166 is not essential for polar localization but rather, it may be involved in the recognition of L-D-cross-linked PG. Consistent with this hypothesis, expression of TtsA$^{N166D}$ did not lead to PG remodeling at the bacterial poles (*Figure 3d and e*). However, introduction of this critical asparagine at an equivalent position in Sen1395 (Sen1395$^{D167N}$) resulted in its partially relocalization to the bacterial poles suggesting a complex role for this residue in both substrate recognition and polar localization (*Figure 3—figure supplement 2*). Taken together, these results implicate a role for the carboxy-terminal domain of TtsA in its polar localization and suggest that polar localization and substrate recognition are functionally separable activities that are mediated by different TtsA determinants.

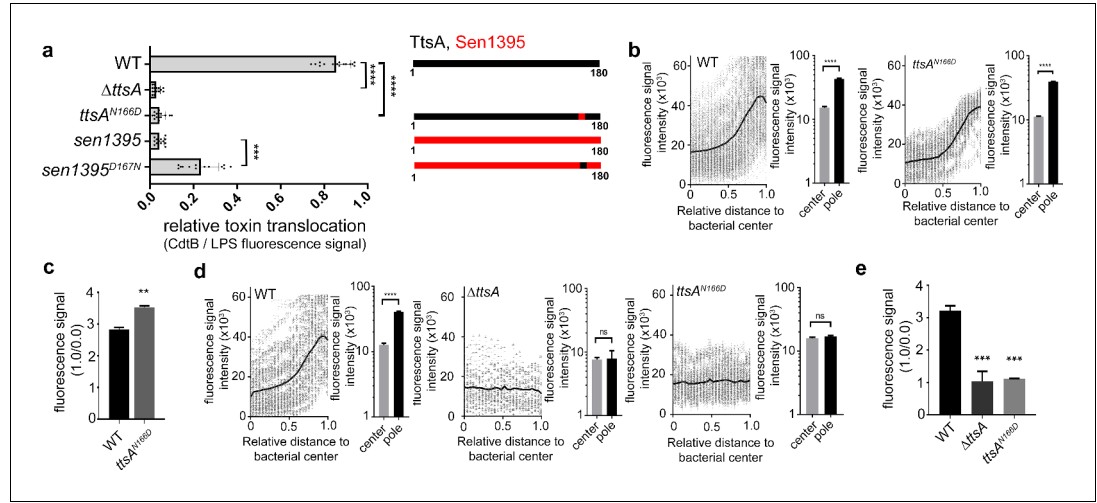

**Figure 3.** Polar localization and substrate recognition are mediated by different TtsA determinants. (a) Typhoid toxin translocation across the PG layer. Wild type *S.* Typhi, the isogenic Δ*ttsA* mutant, or *S.* Typhi strains expressing *ttsA*$^{N166D}$, or *sen1395*$^{D167N}$, all encoding FLAG-tagged CdtB, were grown in TTIM for 24 hr. Bacteria were then fixed, treated with Triton X-100 (0.1%) and stained with mouse anti-FLAG (to stain CdtB) (green fluorescence) and rabbit anti *S.* Typhi LPS (to visualize bacterial cells) (red fluorescence). The relative amount of typhoid toxin translocation across the PG was quantified by immunofluorescence microscopy after staining with antibodies to the FLAG epitope (to visualize CdtB, a component of typhoid toxin), and LPS (to visualize bacterial cells). The average ratios of typhoid toxin- and LPS-associated fluorescence intensity ± standard deviation are shown (****p<0.0001, ***p<0.001, two-sided Student's t-Test). For each experiment a total of 10 images were collected from which 100 randomly selected bacteria per image were analyzed. A diagram of the different constructs is also shown (right panel). (b-c) TtsA subcellular localization in different *S.* Typhi strains. *S.* Typhi strains carrying chromosomally-encoded 3xFLAG-tagged wild type TtsA or TtsA$^{N166D}$ were grown for 24 hr in TTIM, fixed, and stained with a mouse antibody directed to the FLAG-epitope (green) (to visualize TtsA) and a rabbit antibody directed to *S.* Typhi LPS (red). The scatter plot shows the line scan analysis of fluorescence intensity along the axes of individual bacterial cells as described in *Figure 2*. The bar graphs next to each scatter plot show the average ratios of the signal intensities measured at the furthest point from the center (1.0) and at the center of each bacterium (0.0). Data represent the mean ± standard deviation from 1800 measurements (****p<0.0001, two-sided Student's t-Test). (c) Bar graph shows the average ratios between the signal intensity of the indicated proteins measured at the point furthest from the center (1.0) and at the center of each bacterium (0.0). Data represent the mean ± standard deviation (**p<0.01, two-sided Student's t-Test). (d-e) TtsA- and TtsA$^{N166D}$-mediated PG remodeling. *S.* Typhi wild type, the isogenic Δ*ttsA* mutant or a strain expressing *ttsA*$^{N166D}$ were grown in TTIM for 24 hr and the PG was metabolically labeled with alkyne-D-alanine. Remodeling PG was subsequently revealed with azido-AF488 after its linkage to the alkyne-D-alanine that had been incorporated into the PG layer. The scatter plots show the results of line scan analyses of fluorescence signal intensities along the axes of individual bacterial cells as described above. The line depicts the average fluorescence for each measured point. The bar graphs next to the scatter plots show the average ratios of the signal intensities measured at the furthest point from the center (1.0) and at the center of each bacterium (0.0) Data represent the mean ± standard deviation (****p<0.0001, ns p=0.8464, ns p=0.8741 two-sided Student's t-Test) (Note: the data of the Δ*ttsA* mutant is the same as *Figure 2g* and is shown here to facilitate comparison). (e) The average ratios of the signal intensities of the indicated strains measured at the furthest point from the center (1.0) and at the center of each bacterium (0.0). Data represent the mean ± standard deviation (***p<0,001, two-sided Student's t-Test). (a–e) All data are derived from at least three independent experiments (*Figure 3—source data 1*).

The online version of this article includes the following source data and figure supplement(s) for figure 3:

**Source data 1.** Contains source data related to *Figures 3a, b and d*.

**Figure supplement 1.** Ability of *S. Enteritidis* Sen1395-TtsA chimeric or mutant proteins to complement a *S.* Typhi Δ*ttsA* mutant strain for typhoid toxin translocation across the PG.

**Figure supplement 2.** Subcellular localization of Sen1395$^{D167N}$.

## The atomic structure of TtsA reveals two independent domains with distinct functions

To gain insight into the structural organization of TtsA we solved its crystal structure using single wave-length anomalous diffraction (SAD) phasing from a seleno-methionine derivative crystal. The structure was refined to 2.1 Å resolution with an Rwork of 0.19 and an Rfree of 0.22 (*Supplementary file 1*). The crystals belonged to the C 2 2 $2_1$ space group and contained four monomers per asymmetric unit (*Figure 4—figure supplement 1*). This organization does not correspond to a physiological multimer since in solution TtsA behaves as a monomer as determined by size exclusion chromatography. The TtsA structure shows two defined globular α-helical domains separated by a 'hinge' that is formed by a bent α-helix (*Figure 4a*). The amino terminal domain (amino acids 1 to 64) contains the lysozyme-like catalytic triad (Glu14, Asp19, Thr28) situated under a 'flap' formed by a loop that is anchored to the domain by interactions with three short helixes (*Figure 4b*). The carboxy-terminal substrate-binding domain (amino acids 68 to 180) is composed of 6 anti parallel α-helices that organize into a helix bundle (*Figure 4a*). The helix bundle serves as a support for a platform with a central groove, which together with the catalytic triad-containing flap forms a ring-like structure (*Figure 4c*). As will be elaborated below, this ring-like structure is proposed to be the site of engagement of the peptidoglycan substrate. A Dali search with the TtsA structure yielded several structures with significant similarity in the Protein Data Bank, including several amidases and chitosinases. However, two of the structures, NMB1012 from *Neisseria meningitides* (PDB 2is5) and PG_0293 from *Porphyromonas gingivalis* (PDB 2nr7)) shared stronger structural similarity than the others (z scores of 13 and 9.5, respectively). These are proteins of unknown functions whose structures were determined through structural genomics efforts. Although these proteins shared limited primary amino acid sequence similarity with TtsA, their structures aligned well, particularly in some regions of the substrate-binding domain. TtsA has 27% sequence identity with NMB1012 and a root-mean-square deviation (rmsd) of 1.3 Å over 70 aligned Cα pairs, and has 15% sequence identity with PG_0293 and rmsd of 1.0 Å over 34 aligned Cα pairs (*Figure 4—figure supplement 2*). The catalytic domain of *Neisseria meningitides* NMB1012 also aligns well with that of TtsA. In particular, the predicted catalytic residues overlap well in the two structures (*Figure 4—figure supplement 2*). However, in the case of PG_0293, the flap containing the catalytic residues is translocated 90 degrees to the side, placing it away from the proposed substrate-binding site (*Figure 4—figure supplement 2*). The significance of this observation is unclear but this unusual configuration may represent an inactive conformation of an intermediate step in the catalytic reaction, which has been serendipitously captured in the reported crystal.

Electron density not corresponding to protein residues was identified in all four independent TtsA copies in the crystal at one of the ends of what we propose to be the substrate-binding groove of TtsA (see below) (*Figure 4d*). Diaminopimelic acid (API) could be modeled into this density with an optimal fit indicating that this density corresponds to the muropeptide stem (or a portion of it) of an *E. coli* peptidoglycan fragment presumably bound to TtsA after lysis and carried through the purification procedure. The ligand itself sits within a cavity and it is stabilized by hydrogen bond contacts between the NH2 group of API and D118 in TtsA, and the COOH group of API and T125 (*Figure 4d* and (*Figure 4—figure supplement 3*). To assess the potential functional significance of these observations, we introduced a mutation in D118 (TtsA$^{D118A}$) as well as additional amino acids lining the putative substrate-binding groove (TtsA$^{I116D}$, TtsA$^{N166D}$, TtsA$^{Q120R}$, TtsA$^{Q164E}$) and examined the ability of the resulting mutants to complement a ΔttsA *S.* Typhi mutant for its ability to translocate typhoid toxin across the PG layer. We found that S. Typhi expressing TtsA$^{D118A}$, TtsA$^{I116D}$, or TtsA$^{N166D}$ were unable to translocate typhoid toxin indicating that these residues play a critical role in substrate recognition (*Figure 4e* and *Figure 4—figure supplement 4*).

Comparison of the structures of the different molecules within the TtsA asymmetric unit revealed two different conformations, one conformation shared by chains A, B, and D, and another conformation in chain C. These two alternative conformations, while overall largely identical, position the flap that contains the catalytic residues either closer or further apart (~6.1 Å) from the substrate-binding groove, resulting in either a close or open ring-like structure (*Figure 4f* and *Video 1*). The repositioning of the catalytic site in these two configurations is the consequence of discrete conformational changes in the hinge that separates the amino terminal (catalytic) and carboxy-terminal (substrate-binding) domains. Although the functional relevance of these two conformations is unknown, it is

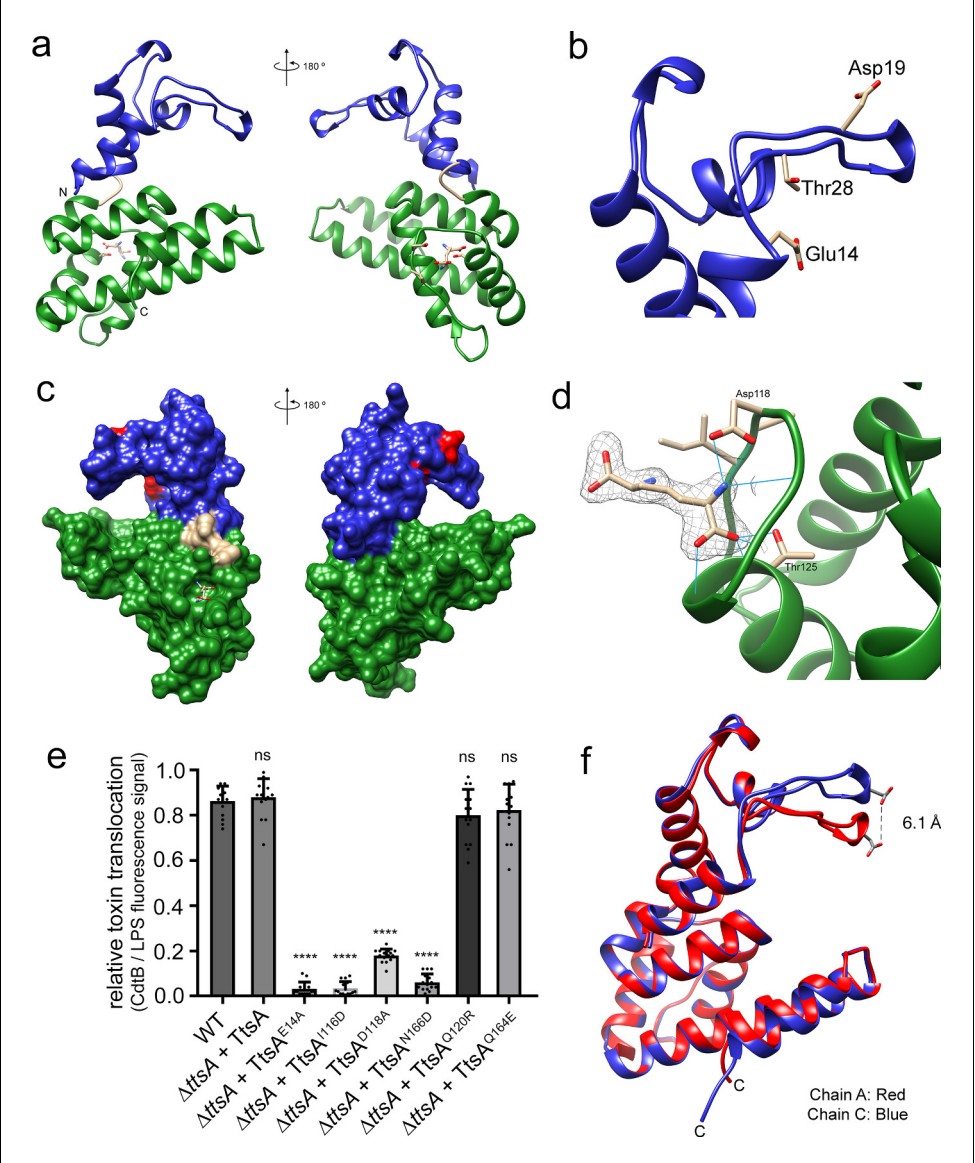

**Figure 4.** The atomic structure of TtsA reveals two independent domains with distinct functions. (**a**) Two opposite views of the overall structure of TtsA in ribbon representation. The catalytic domain is shown in blue while the substrate-binding domain is shown in green, linked by a loop (cream colored). (**b**) Close up view of the catalytic domain showing the lysozyme-like catalytic triad in TtsA. (**c**) Two opposite views of the surface rendering of TtsA. The color scheme is the same as in panel (**a**) with the catalytic triad colored in red. (**d**) Close up view of the diaminopimelic acid (API) binding pocket in the substrate-binding domain. The electron density map (2Fo-Fc, 1σ level) is shown with the modeled API (DAP) and hydrogen bonds between API (DAP) and TtsA marked by blue lines. (**e**) Typhoid toxin translocation across the PG layer in *S.* Typhi strains expressing structurally-guided TtsA mutants. *S.* Typhi wild type and isogenic Δ*ttsA* mutant strains (all expressing 3xFLAG-tagged CdtB) carrying plasmids expressing from an arabinose-inducible promoter the indicated *ttsA* mutants, were grown in TTIM containing 0.001% arabinose for 24 hr at 37°C. The relative amount of typhoid toxin translocation across the PG was quantified by immunofluorescence microscopy after staining with antibodies to the FLAG epitope (to visualize CdtB, a component of typhoid toxin), and LPS (to visualize bacterial cells). The average ratios of typhoid toxin- and LPS-associated fluorescence intensity ± standard deviation are shown (****p<0.0001, ns p=0.5629, p=0.0749, p=0.2450, two-sided Student's t-Test). For each experiment a total of 10 images were collected from which 100 randomly selected bacteria per image were analyzed. Data are derived from three independent experiments (*Figure 4—source data 1*). (**f**) Comparison of the crystal structures of the TtsA chain A in red and chain C in blue, showing the conformational differences in the catalytic flap.

The online version of this article includes the following source data and figure supplement(s) for figure 4:

*Figure 4 continued*

**Source data 1.** Contains source data related to *Figure 4e*.
**Figure supplement 1.** Overall view of the four monomers present in the asymmetric unit of the TtsA crystal.
**Figure supplement 2.** Structural comparison of TtsA with NMB1012 from *Neisseria meningitides* (PDB 2is5, left panel) and PG_0293 from *Porphyromonas gingivalis* (PDB 2nr7, right panel).
**Figure supplement 3.** Close up of the surface rendering of the diaminopimelic (API) binding pocket.
**Figure supplement 4.** Western blot analysis of the expression levels of TtsA and CdtB in the indicated *S. Typhi* strains assayed in *Figure 4E*.

possible that they represent alternative conformations adopted by TtsA during catalysis. Taken together, the atomic structure of TtsA revealed a defined domain organization and provided major insight into the structural bases for substrate engagement.

## Structural bases for TtsA substrate specificity

As shown above, Sen1395, a close TtsA homolog from *S. Enteritidis*, is unable to complement a Δ*ttsA S.* Typhi mutant for typhoid toxin translocation because it exhibits different substrate specificity. Through the analysis of different TtsA/Sen1395 chimeras we were able to determine that the inability of Sen1395 to complement a Δ*ttsA* mutant is due to differences in the substrate-binding domain. To gain insight into the mechanisms of TtsA substrate specificity, we solved the atomic structure of Sen1395 to 1.9 Å resolution with an Rwork of 0.20 and an Rfree of 0.24 (*Supplementary file 1*). The crystals belonged to the P1 space group and contained two monomers per asymmetric unit (*Figure 5—figure supplement 1*). The structure aligns very well to TtsA with an rmsd of ~0.7 Å over 161 aligned Cα pairs (with 64% sequence identity) (*Figure 5a*). The positions of the conserved catalytic residues in both enzymes overlap reasonably well, which is consistent with the observation that the catalytic domain of TtsA can be replaced with the catalytic domain of Sen1395 and retain full function. Despite the very high structural similarity of the two enzymes, notable differences on amino acids located on the surface of the putative substrate-biding groove could be identified (*Figure 5b* and *Figure 5—figure supplement 2*). Notably, while the surface of the substrate-binding groove in TtsA is lined by largely uncharged amino acids (Ile116, Gln120, Gln164, Asn166), the equivalent surface in Sen1395 is occupied by several charged amino acids (Asp117, Arg121, Glu165, Asp167). We propose that the charge differences in the substrate-binding domain between TtsA and Sen1395 confer upon these enzymes their different substrate specificity.

To more directly probe the importance of the putative substrate-binding groove in TtsA we placed a photoreactive p-benzoyl-L-phenylalanine (pBpa) amino acid (*Lee et al., 2009*) at critical sites lining this region by introducing amber (TAG) codons in the positions encoding D118, N166, and F169. We also placed a TAG codon at the position encoding the catalytic residue E14, which is located away from the substrate-binding groove. A plasmid encoding orthogonal suppressor aminoacyl-tRNA synthetase/tRNA pair, which can recognize amber codons and insert pBpa at these positions, was introduced in all the mutant strains. Bacteria were then grown for 24 hr in TTIM under conditions that are permissive for TtsA expression and typhoid toxin secretion, exposed to UV light, and cross-linked species were analyzed by western blotting. Without UV light exposure all constructs showed similar protein expression levels (*Figure 5c*). However, after exposure to UV light, TtsA^D118pBpA^, TtsA^N166pBpA^ and TtsA^F169pBpA^ were no longer detectable by western blot, although TtsA^E14pBpA^ was detectable at equal levels than those prior to UV exposure (*Figure 5c*). We hypothesized that the inability to detect these TtsA constructs was due to their crosslinking to the peptide stem of the PG polymer, which because of its high molecular weight prevented their migration into SDS-

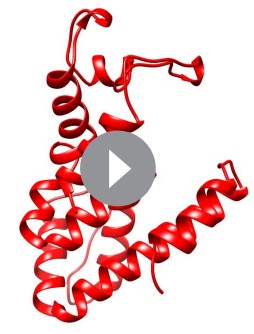

**Video 1.** Conformational differences between chain A and chain C of the TtsA crystal.
https://elifesciences.org/articles/53473#video1

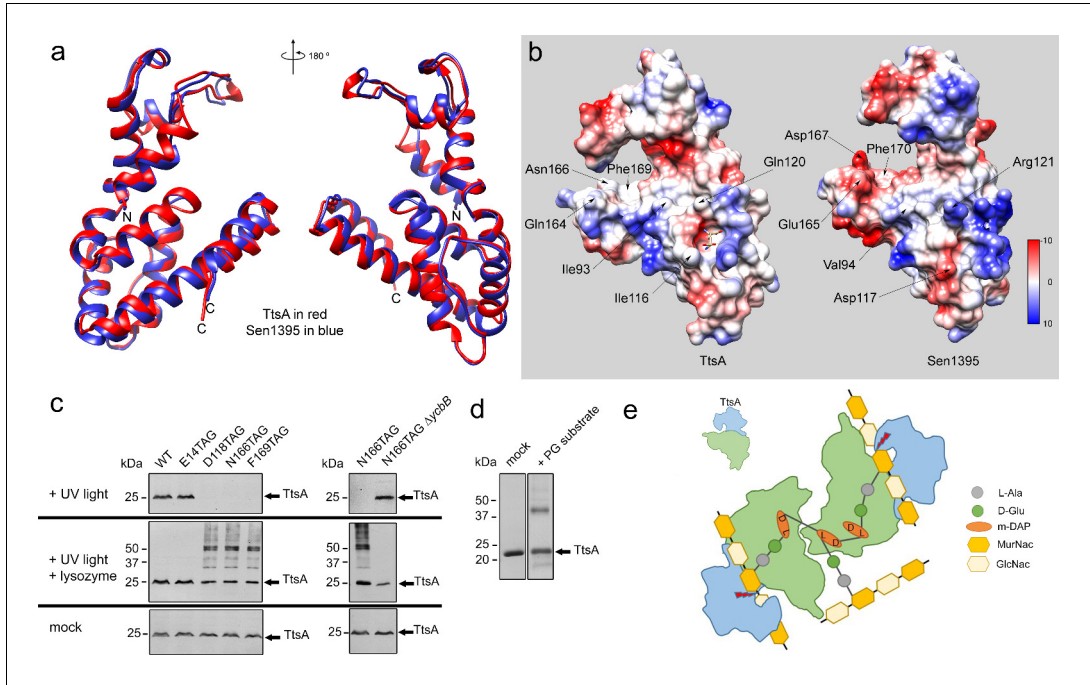

**Figure 5.** Structural bases for TtsA substrate specificity. (a) Two opposite views of the comparison of the crystal structures of TtsA (in red) and Sen1395 (in blue) in ribbon representation. (b) Surface rendering of TtsA and Sen1395 colored by electrostatic potential according to Coulomb's law (blue: positive; red: negative). Critical residues in the substrate-binding groove are indicated. (c) In-vivo photo-crosslinking of selected TtsA residues within its substrate-binding groove. Wild-type *S.* Typhi, an isogenic *ycbB* mutant, or *S.* Typhi strains carrying chromosomally encoded 3xFLAG-tagged wild type and mutant TtsA with an amber codon (TAG) at positions E14, D118, N166 or F169, all expressing the orthogonal suppressor aminoacyl-tRNA synthetase/tRNA pair, were grown 24 hr at 37°C in TTIM containing 1 mM pBpa. Cultures were split and exposed to UV light or exposed to UV light followed by treatment with lysozyme (as indicated). Cross-linked species were analyzed by western blotting using mouse-anti FLAG antibodies. (d) In vitro chemical crosslinking of purified TtsA. Purified TtsA was mixed with PBS (mock) or mixed with purified PG isolated from *S.* Typhi. The mixtures were incubated with the amine-to-amine crosslinker $BS_3$ (bis(sulfosusinimidyl)suberate) and subsequently analyzed by SDS-PAGE. (e) Model for the TtsA mechanism of substrate recognition. TtsA monomers recognize adjacent peptide stems of PG polymers that are the subject of complex L-D crosslinking patterns that yield trimers and tetramers after lysozyme treatment. The cartoon does not imply any specific position of any specific domain of the TtsA molecule relative to the stem peptides or the glycan strands.

The online version of this article includes the following figure supplement(s) for figure 5:

**Figure supplement 1.** Overall view of the two monomers present in the asymmetric unit of the Sen1395 crystal.
**Figure supplement 2.** Critical non-conserved residues in the substrate-binding groove of TtsA and Sen1395.

PAGE gels. To test this hypothesis, we treated the crosslinked samples with lysozyme, which digests PG into its muropeptide subunits and analyzed the digested samples in SDS-PAGE probing for TtsA. We found that after treatment with lysozyme we were able to readily detect TtsA$^{D118pBpA}$, TtsA$^{N166pBpA}$ and TtsA$^{F169pBpA}$ at equal levels than TtsA$^{E14pBpA}$ Indicating that these mutant constructs were crosslinked to PG after UV treatment (*Figure 5c*). Interestingly, in the case of TtsA$^{D118pBpA}$, TtsA$^{N166pBpA}$ and TtsA$^{F169pBpA}$ we detected two bands after lysozyme treatment: one migrating as ~25 kDa polypeptide, the predicted size of a TtsA monomer, and another migrating as a 50 kDa polypeptide, consistent with the size of a TtsA dimer (*Figure 5c*). Since TtsA behaves as a monomer in solution, the dimer is unlikely to be the result of TtsA/TtsA crosslinks. Furthermore, the location of the labeled amino acids in the predicted substrate-binding groove within the TtsA structure dictates that it is unlikely that this region could engage into putative TtsA-TtsA interactions. Rather, these results suggest that TtsA must bind two adjacent cross-linked muropeptides. Consistent with this hypothesis, only monomers (with or without lysozyme treatment) were detected when the experiment was conducted in a *S.* Typhi Δ*ycbB* strain, which lacks LD-crosslinked PG and

therefore it is not permissive for TtsA activity (*Figure 5c*). To further explore this hypothesis we performed in vitro chemical crosslinking experiments using bis(sulfosusinimidyl)suberate (BS3), an amine-to-amine crosslinker with an 8-carbon spacer arm. Purified TtsA, and PG obtained from *S.* Typhi grown in TTIM were incubated with the crosslinker and subsequently analyzed by SDS-PAGE (*Figure 5d*). Consistent with our observation that purified TtsA behaves as a monomer in solution, addition of the crosslinker to purified TtsA did not result in the formation of a dimer. In contrast, addition of the crosslinker to purified TtsA in the presence of purified peptidoglycan resulted in the formation of a TtsA species that migrated at ~50 kDa. Since the efficiency of both the in vitro and in vivo crosslinking reactions is expected to be much less than 100%, the finding of a substantial proportion of crosslinked dimeric TtsA is consistent with the notion that two TtsA molecules preferably bind two adjacent, cross-linked muropeptides.

## Discussion

The secretion of typhoid toxin is dependent on a unique protein secretion mechanism specifically adapted for the biology of the intracellular pathogen *S.* Typhi (*Geiger et al., 2018*). In this secretion mechanism, the typhoid toxin subunits, PltA, PltB, and CdtB, are first secreted to the periplasmic space by the *sec* translocation machinery through the engagement of canonical secretion signals in each one of these polypeptides. Once in the periplasm, the protein subunits are assembled into the holotoxin complex, which is subsequently translocated to the *trans* side of the PG layer. From this location, the holotoxin is finally released to the outside after exposure to membrane-active compounds such as antimicrobial peptides or bile salts, which are encountered by the bacteria during infection and can introduce minor disruptions to the outer membrane. The critical step of toxin translocation through the PG layer occurs at the bacterial poles and is mediated by TtsA, which belongs to a unique family of muramidases most often associated with bacteriophage release (*Hodak and Galán, 2013*; *Stojković and Rothman-Denes, 2007*; *Pei and Grishin, 2005*). Unlike its bacteriophage homologs, however, TtsA exhibits exquisite sensitivity for L-D-crosslinked PG, which in most Gram-negative bacteria are less abundant than the most commonly found D-D crosslinked species. Here we have investigated the mechanisms by which TtsA is recruited to the bacterial poles and specifically targets L-D crosslinked PG. By constructing chimeras with the *S.* Enteritidis Sen1395 protein, a close homolog of TtsA with different specificity and subcellular localization, we were able to map the substrate specificity and subcellular localization determinants to its carboxy terminal domain. This finding raised the possibility that the presence of L-D crosslinked PG itself may serve as a signal to target TtsA to the poles, since we have found that YcbB, which in *S.* Typhi is the only enzyme capable of introducing this modification, is also enriched at the poles. However, we found that TtsA also localized to the poles of a *S.* Typhi Δ*ycbB* mutant, which lacks L-D crosslinked PG, although with reduced efficiency. These observations suggest that polar localization can occur without this PG modification and that other determinants must contribute to the targeting of TtsA to this subcellular location. These results also indicate that the mechanisms of polar localization and L-D PG targeting are functionally distinct.

To gain insight into the structural bases for substrate specificity, we solved the crystal structures of TtsA and its close homolog Sen1395, which exhibits different specificity. Entirely consistent with the chimera studies, we found that TtsA is organized in two independent domains: an amino-terminal catalytic domain and a carboxy-terminal substrate-binding domain, separated by a bent helix that forms a hinge between these two domains. The conserved, lysozyme-like catalytic triad is located within a loop that is supported by three short helixes. This loop reaches out to the substrate-binding groove and in doing so forms a ring-like structure that presumably envelopes the PG and positions the catalytic residues for the cleavage of the PG backbone. The carboxy-terminal domain contains a helix bundle that forms a platform with a central groove that we postulate serves as a binding site for the muropeptide stem. In support of this conclusion, we observed electron density corresponding to diaminopimelic acid bound to a crevice on one of the ends of the substrate-binding groove. Mutations in the residue that coordinates the binding to diaminopimelic acid resulted in complete loss of function, consistent with the notion that this residue plays an essential role in substrate recognition. Comparison of the TtsA and Sen1395 atomic structures revealed specific differences in critical amino acids lining the substrate-binding groove, which result in defined

charge differences lining the substrate-binding groove. We postulate that these charge differences are responsible for the differences in substrate specificity.

To further evaluate the functional importance of residues lining the predicted substrate-binding groove of TtsA, we introduced the genetically-encoded photo-crosslinkable amino acid pBpa. We then examined the mobility of these mutant proteins after *S.* Typhi growth under conditions permissive for typhoid toxin secretion and exposure to UV light. We found that mutants containing pBpa at positions D118, N166, and F169 cross-linked to PG after exposure to UV light. Interestingly, a significant proportion of the TtsA cross-linked species migrated in SDS-PAGE as dimers. TtsA does not form dimers in solution therefore we interpret these observations to mean that the two monomers must be held together by cross-linked PG species rather than by direct TtsA-TtsA crosslinks. Consistent with this hypothesis, TtsA exposure to a chemical crosslinker in solution yielded dimers only in the presence of PG. As the TtsA dimers resisted lysozyme treatment, we hypothesize that the TtsA monomers must be cross-linked to adjacent peptide stems of the PG polymer (*Figure 5e*). These peptide stems must be the subject of complex crosslinking patterns since lysozyme-treated PG with canonical LD-crosslinks should yield unlinked monomers, which in turn would be incompatible with the observed TtsA dimers after UV exposure. LD transpeptidation does not require de novo incorporation of new peptidoglycan precursors, therefore it has been observed that the proportion of LD-crosslinked peptide increases in stationary phase (*Turner et al., 2014*; *de Pedro and Cava, 2015*; *Typas et al., 2012*). We have shown that the only LD-transpeptidase in *S.* Typhi, YcbB, localizes to the bacterial poles so it is possible that the presence of this enzyme at that location may result in the accumulation of cross-linked trimers and tetramers that may be selectively targeted by TtsA (*Figure 5e*). This in turn may provide a mechanism to restrict the enzymatic activity of TtsA to the bacterial poles.

Comparison of the structures of the four independent molecules in the TtsA crystal revealed the presence of two conformations: an open conformation in which the loop containing the catalytic residues is moved away from the substrate, and a close conformation with the loop in much closer proximity to the substrate-binding groove. Similarly, comparison of the TtsA structure with those available in the PDB data-base revealed close similarity to structures from *Neisseria meningitides* (NMB1012) and *Porphiromonas gingivalis* (PG_0293). There is no functional information available about these proteins as their structures were obtained through structural genomics efforts, therefore their PG specificity is unknown and it is unclear whether they exert their function in the context of phage biology or protein secretion. Intriguingly, however, the catalytic loop of PG_0293 is rotated ~180° to the side so that it is positioned away from the substrate-binding domain and presumably unable to make contact with its substrate. Although the significance of the observed alternative conformations is unclear, it is tempting to hypothesize that they may represent functionally relevant conformational states serendipitously captured in the crystal that are perhaps associated with substrate release after catalysis.

In summary, using a multidisciplinary approach we have provided major insight into the mechanisms of substrate specificity exhibited by a unique bacterial muramidase associated with the secretion of typhoid toxin, a major virulence factor of the human pathogen *S.* Typhi. These studies may therefore provide the bases for the potential development of anti-toxin strategies.

# Materials and methods

## Key resources table

| Reagent type (species) or resource | Designation | Source or reference | Identifiers | Additional information |
|---|---|---|---|---|
| Gene (*Salmonella enterica* serovar Typhi) | *ttsA* | PMID: 23174673; MicrobesOnline Database | sty1889 | |
| Gene (S. Typhi) | *ycbB* | PMID: 30250245; MicrobesOnline Database | sty0997 | |

*Continued on next page*

*Continued*

| Reagent type (species) or resource | Designation | Source or reference | Identifiers | Additional information |
|---|---|---|---|---|
| Gene (*Salmonella enteritidis*) | *sen1395* | PMID: 30250245; MicrobesOnline Database | sen1395 | |
| Gene (S. Typhi) | *cdtB* | PMID: 18191792; MicrobesOnline Database | sty1886 | |
| Strain, strain background (*Escherichia coli*) | BL21 (DE3) | Invitrogen | | Galan lab; electro competent |
| Strain, strain background (*Escherichia coli*) | β−2163 Δ*nic* 35 (DAP-) | PMID: 15748991 | | |
| Strain, strain background (*Escherichia coli*) | B834 (DE3) | Novagen | Cat. No. 69041 | |
| Strain, strain background (*Salmonella enterica serovar Typhi*) | ISP2825 | PMID: 1879916 | | |
| Antibody | anti-FLAG M2 (Monoclonal mouse) | Sigma-Aldrich | | 1:10000 |
| Antibody | anti-Salmonella O poly A-1 and Vi (Polyclonal rabbit) | Becton, Dickinson and Co | | 1:10000 |
| Antibody | Alexa-Fluor 488-conjugated anti-mouse | Invitrogen | | 1:2000 |
| Antibody | Alexa-Fluor 594-conjugated anti-rabbit | Invitrogen | | 1:2000 |
| Recombinant DNA reagent | pSB 890 (plasmid) | PMID: 7997169 | | |
| Recombinant DNA reagent | pET28b (plasmid) | Invitrogen | | |
| Recombinant DNA reagent | pSB 3783 (plasmid) | PMID: 30250245 | pBAD24 | |
| Sequence-based reagent | ttsAfor | This paper | PCR primers | ATGACTAAAGATGAAATC |
| Sequence-based reagent | ttsArev | This paper | PCR primers | TTACAATCTTACCCGTTC |
| Sequence-based reagent | sen1395for | This paper | PCR primers | ATGAAACCGAAGGACGAA |
| Sequence-based reagent | sen1395rev | This paper | PCR primers | TCATATCAATACGCGCTC |
| Sequence-based reagent | ycbBfor | This paper | PCR primers | ATGTTGCTTAATAAGATG |
| Sequence-based reagent | ycbBrev | This paper | PCR primers | TTACCTGATTAATTGTTC |
| Software, algorithm | ImageJ with Microbe J plug in | http://rsbweb.nih.gov/ij/; PMID: 27572972 | | |

## Bacterial strains and plasmids

The bacterial strains and plasmids used in this study are listed in *Supplementary file 2*. All *S.* Typhi strains are derived from the clinical isolate ISP2825 (*Galán and Curtiss, 1991*). All in frame deletions or insertions into the *S.* Typhi chromosome were generated by standard recombinant DNA and allelic exchange procedures using the *E. coli* β−2163 Δnic35 as the conjugative donor strain (*Demarre et al., 2005*) and the R6K-derived suicide vector pSB890 as previously described (*Kaniga et al., 1994*). For *S.* Typhi Δ*ttsA* complementation studies, we used plasmid pSB3783, which encodes an arabinose-inducible promoter and is derived from plasmid pBAD24 (*Guzman et al., 1995*). For TtsA and Sen1395 protein expression and purification, we used the expression plasmid pET28b (Novagen) in *E. coli* strain BL21. The methionine auxotrophic *E. coli* strain B834 (DE3) (Novagen) carrying a plasmid derived from pET28b and encoding *ttsA* was used for selenomethionine metabolic labeling of TtsA. All plasmids used in this study were constructed using the Gibson assembly cloning strategy (*Gibson et al., 2009*). All generated plasmids and strains used in this study have been verified by nucleotide sequencing.

## Bacterial cultures

*S.* Typhi strains were routinely cultured in L-broth (LB) on a rotation wheel at 37°C. For in vitro typhoid toxin secretion assays, bacteria were sub-cultured in a chemically defined medium, which is permissive for typhoid toxin expression and secretion and therefore is referred to as TTIM for typhoid toxin expression inducing medium. This defined medium was adapted from previous studies (*Fowler and Galán, 2018*; *Beuzón et al., 1999*) and its composition is as follows: $K_2SO_4$ (0.5 mM), $KH_2PO_4$ (1 mM), $(NH_4)_2SO_4$ (7.5 mM), Tris Base (50 mM), Bis Tris (50 mM), casamino acids (0.1%), KCl (5 mM), cysteine (50 µg/ml), tryptophan (50 µg/ml), glycerol (32.5 mM) and magnesium (15 µM). When appropriate, ampicillin (100 µg/ml), kanamycin (50 µg/ml) and tetracycline (10 µg/ml) were added to the bacterial cultures.

## Detection of TtsA, Sen1395 and chimeric protein expression in vitro

To detect the expression of typhoid toxin and various muramidases, *S.* Typhi strains carrying chromosomally encoded 3xFLAG-epitope-tagged versions of CdtB (a typhoid toxin subunit), various 3xFLAG-epitope-tagged muramidases (as indicated), or plasmid-born 3xFLAG-epitope-tagged muramidases were grown overnight in LB [when appropriate with addition of tetracycline (10 µg/ml)], washed twice with 1 x PBS (without $MgCl_2$ supplement, Difco), and sub-cultured in TTIM after a 1:50 dilution. At the indicated time points, an equal number of bacteria (standardized by $OD_{600}$ measurements) were harvested and bacterial pellets were resuspended in Laemmli buffer and boiled for 5 min. The expression profile of the different muramidases and CdtB was determined by Western blot analysis using a mouse monoclonal antibody directed to the FLAG-epitope (Invitrogen). As a loading control, samples were analyzed on a separate Western blot with a polyclonal rabbit antibody directed to RecA.

## Quantification of typhoid toxin translocation across the PG by fluorescence microscopy

To quantify the amount of typhoid toxin translocated through the PG layer, we used a previously described protocol (*Geiger et al., 2018*). Briefly, the indicated *S.* Typhi strains were grown in TTIM for 24 hr and bacteria were then washed with PBS and fixed with 4% paraformaldehyde (PFA) for 15 min. Bacteria were then washed and mounted on poly-(D)-lysine (Sigma-Aldrich) coated glass coverslips. The coverslips with attached bacteria were washed with PBS and incubated with 1% Triton X-100 in PBS for 30 min, washed again, and incubated overnight at 4°C with primary anti-FLAG M2 mouse monoclonal antibody (Sigma) (1:10,000) and anti-Salmonella O poly A-1 and Vi rabbit antiserum (Becton, Dickinson and Co.) (1:10,000) in PBS, containing 1% bovine serum albumin (BSA) and 0.01% Triton X-100. After removal of the primary antibody, the coverslips were washed 6 times with 1 x PBS and incubated with secondary Alexa-Fluor 488-conjugated anti-mouse and Alexa-Fluor 594-conjugated anti-rabbit (Invitrogen) (1:2,000) antibodies in PBS, containing 1% bovine serum albumin (BSA) and 0.01% Triton X-100 for 60 min at RT protected from light. Mounted samples (ProLong Gold antifade, Molecular Probes) were visualized in an Eclipse TE2000-U (Nikon) microscope equipped with an Andor Zyla 5.5 sCMOS camera driven by Micromanager software (https://www.

micro-manager.org). Bacteria-associated fluorescence signal from FLAG-tagged CdtB (to visualize typhoid toxin), were quantified by fluorescence microscopy using the open source software ImageJ (http://rsbweb.nih.gov/ij/) with the MicrobeJ plug-in (*Ducret et al., 2016*). Images were captured from randomly selected fields in both, the red (LPS signal to identify all bacterial cells) and green (to identify CdtB positive bacteria) channels. The total signal intensities of bacterial cells positive for green fluorescence signals (CdtB) was measured and quantified relative to the total signal intensities of bacterial cells analyzed (LPS signal, red). For each experiment a total of 30 images were collected from which 100 randomly selected bacteria per image were analyzed resulting in 3000 bacterial cells analyzed per bacterial strain and/or condition. All experiments were repeated three times, independently.

## Fluorescence labeling of peptidoglycan

The indicated *S.* Typhi strains were grown in TTIM for 24 hr, and subsequently incubated in TTIM containing alkyne-D-alanine (2 mM) (Boao Pharma, Boston) for 4 hr. Bacteria were washed with PBS and fixed for 15 min in 4% paraformaldehyde (PFA) at room temperature. Fixed bacteria were resuspended in Click-iT Cell Reaction Buffer (Invitrogen) and copper-catalyzed click-chemistry was performed in the dark at room temperature for 60 min using the Click-iT Cell Reaction Kit (Invitrogen) with 10 µM azido-Alexa-Fluoro488 fluorophore (Invitrogen). Subsequently, bacteria were washed three times and attached to poly-(D) lysine coated coverslips. Bacterial cells were counterstained for LPS (red) with primary anti-*Salmonella* O poly A-1 and Vi rabbit antiserum (Becton, Dickinson and Co.) (1:10,000) and secondary anti-rabbit Alexa-Fluor594 (Invitrogen) (1:2,000) antibody.

## Fluorescence distribution analyses for bacterial proteins and labeled peptidoglycan

The line scan analysis for the distribution of fluorescence signal intensities of labeled peptidoglycan or 3xFLAG-tagged proteins along the median longitudinal axes of the bacterial cell bodies was carried out using the MicrobeJ plug-in (*Ducret et al., 2016*) of the ImageJ software (https://imagej.nih.gov/ij/index.html) as previously described (*Geiger et al., 2018*). Briefly, random fluorescence microscopy images were obtained and bacterial cells were defined as regions of interest (ROI) by using LPS (red) counterstaining. The center within each ROI was identified and 26 measuring points along the median longitudinal axes of the bacteria from the center (0) to each of the poles (1.0) were automatically selected on the peptidoglycan or labeled antibody fluorescence channels (green) for protein detection. For the peptidoglycan-associated fluorescence distribution analysis, the fluorescence signal associated with non-TtsA-dependent PG labeling was substracted.

## TtsA and Sen1395 protein purification

The coding sequence for *ttsA* and *sen1395* was amplified from *S.* Typhi or *S. enteritidis* strains, respectively, and cloned by Gibson cloning strategy (*Gibson et al., 2009*) into the expression vector pET28b (Novagen) resulting in N-terminal histidine-epitope tagged TtsA or Sen1395. *Escherichia coli* strain BL21 carrying the different plasmids were grown in LB containing kanamycin (50 µg/ml) to an $OD_{600\ nm}$ of 0.6–0.7 at 37°C. Expression of TtsA and Sen1395 was subsequently induced by the addition of 0.5 mM IPTG, and induced cultures were incubated overnight at 25°C. Bacterial cells were pelleted by centrifugation, resuspended in lysis buffer [20 mM Tris-HCl; pH 8, 150 mM NaCl, lysozyme (100 µg/ml), DNAse (100 µg/ml), saturated PMSF] and lysed with a high pressure cell disruptor (Constant Systems Ltd.). For Sen1395 a protein denaturation/renaturation protocol was used to obtain soluble protein from inclusion bodies. More specifically, the cell lysate was clarified by centrifugation (20,000 x g for 1 hr at 4°C) to recover the pellet containing the cellular debris and inclusion bodies. Inclusion bodies were solubilized by resuspension in solubilization buffer (8 M urea, 100 mM $NaH_2PO_4$, 10 mM Tris; pH 8) and the suspension was stirred at room temperature for 3 hr. After incubation, the solubilized material was clarified by centrifugation (20,000 x g for 1 hr at 4°C) and the supernatant containing the solubilized His-tagged Sen1395 protein, was affinity-purified using a Nickel-NTA agarose (Qiagen) column. The column was washed five times with wash solution (8 M urea, 100 mM $NaH_2PO_4$, 10 mM Tris; pH 6.5), protein eluted with elution buffer (8 M urea, 100 mM $NaH_2PO_4$, 100 mM EDTA, 10 mM Tris; pH 4.5), and refolded by dialysis against a renaturalization buffer (25 mM Tris; pH 8, 200 mM NaCl, 10% glycerol, 2 mM EDTA, 5 mM DTT) overnight at 4°C.

Precipitated debris/misfolded proteins were removed by centrifugation (20,000 x g, 30 min at 4°C) and the clear supernatant was used in further purification steps. For TtsA, the cells lysate was pelleted (20,000 x g 1 hr at 4°C) and affinity-purified using a Nickel-NTA agarose (Qiagen) column. Renatured Sen1395 and TtsA were diluted (1:6) in 20 mM Tris-HCl, pH 8.0 buffer and loaded onto a Hi Trap Q ion-exchange column (GE Healthcare). Fractions from the ion-exchange chromatography were monitored on SDS–PAGE, concentrated, and further purified over a Superdex 200 10/300 GL gel filtration column (GE Healthcare). Final fractions were examined for purity on 12% SDS–PAGEs, aliquoted and frozen at −80°C in storage buffer (20 mM Tris; pH 8, 150 mM NaCl).

### Selenomethionine-labeling

The methionine auxotrophic *E. coli* strain B834 (DE3) (Novagen Cat. No. 69041) expressing TtsA (N-terminal His-tag) on a pET28b plasmid was grown over night at 37°C in M9 medium containing kanamycin (50 µg/ml) and methionine (50 µg/ml). The following day bacteria were sub-cultured (1:100 dilution) in M9 medium containing kanamycin (50 µg/ml) and methionine (50 µg/ml) and incubated to an $OD_{600 \ nm}$ of 0.7 at 37°C. The bacteria were pelleted and washed twice with 1xPBS to remove residual methionine. Then the bacteria pellet was resuspended in the same volume of M9 medium without methionine and further incubated until bacterial growth completely stops (~4–5 hr). Seleno-methionine (50 µg/ml) and IPTG (0.5 mM) were then added and the bacterial culture was further incubated at 25°C for 18 hr. Bacterial cells were pelleted by centrifugation, resuspended in lysis buffer [20 mM Tris-HCl; pH 8, 150 mM NaCl, lysozyme (100 µg/ml)], DNAse I (100 µg/ml), saturated PMSF) and further processed as indicated above.

### Peptidoglycan hydrolysis assays

Wild-type *S.* Typhi, its isogenic Δ*ycbB* mutant, or *S.* Typhi wild type harboring an arabinose inducible plasmid expressing *ycbB* were grown in TTIM for 24 hr (if applicable with addition of tetracycline 10 µg/ml) or in LB to logarithmic growth phase ($OD_{600}$ 0.5), and peptidoglycan was isolated as previously described (*Heidrich et al., 2001*). Purified peptidoglycan preparations were resuspended in MilliQ water containing 0.01% $NaN_3$ and stored at 4°C until their use in different assays. Muramidase activity of purified TtsA or Sen1395 was detected by a turbidimetry assay or by dye-release assay on Remazol Brilliant Blue (RBB)-labeled peptidoglycan (*Heidrich et al., 2001*). For the turbidimetric assay isolated peptidoglycan was diluted in enzyme activity buffer (20 mM Tris; pH6.8, 50 mM NaCl) to an $OD_{540}$ of ~0.175. Purified TtsA (5 µg) and Sen1395 (5 µg) were added and optical density measurements at 540 nm were conducted at the indicated time points. In this assay, insoluble peptidoglycan is cleaved into small soluble PG fragments (i.e. muropeptides) leading to a decline in the optical density. For the Remazol Brilliant Blue (RBB)-dye release assay isolated PG was labeled by incubating it in 20 mM RBB (Sigma) in 0.25 M NaOH at 37°C overnight. The sample was neutralized with HCl, centrifuged (20,000 g, 15 min at room temperature), and the resulting pellet was thoroughly washed with MilliQ water until no soluble RBB could be detected after centrifugation. For PG hydrolysis activity tests, 20 µl of indicated RBB-PG was incubated with purified TtsA (5 µg) or Sen1395 (5 µg) in enzyme activity buffer (20 mM Tris; pH6.8, 50 mM NaCl) for 4 hr in a total volume of 100 µl. The samples were centrifuged for 10 min at 20,000 g and the absorbance of the supernatant was measured at 595 nm (Tecan Infinite 200 PRO microplate reader, Tecan Austria GmbH, Austria). Lysozyme (5 µg) and bovine serum albumin (BSA, 5 µg) were used in the assay as positive and negative controls, respectively.

### Bacterial growth arrest assay

*S.* Typhi wild-type and Δ*ycbB* mutant strains, expressing plasmid-born epitope-tagged TtsA, Sen1395 or chimeric proteins under the control of an arabinose inducible promoter, were grown overnight in LB medium supplemented with 10 µg/ml tetracycline at 37°C. Overnight grown bacteria were subcultured (1:50) in TTIM and grown to an $OD_{600}$ of 0.3, at which point 0.3% arabinose was added to the bacterial cultures to induce the expression of the different muramidases, and subsequently incubated for 20 hr at 37°C. Colony forming units (CFUs) were determined by plating bacterial dilutions on LB agar plates.

## In-vitro crosslinking analyses

Purified TtsA (10 μg), was mixed with purified PG isolated from *S.* Typhi that had been grown in TTIM for 24 hr (conditions permissive for typhoid toxin secretion) in reaction buffer (100 mM sodium phosphate, 150 mM NaCl, 20 mM HEPES; pH 8) containing the BS$_3$ crosslinker (Thermo Scientific), either 200 (20x) or 500 μg (50 x) from a stock solution (3 mg/ml) prepared immediately before use. Samples were incubated at room temperature for 30 min and the crosslinking was quenched by the addition of quenching buffer (200 mM Tris; pH 7.5) for 15 min at room temperature. Samples were resuspended in SDS loading buffer, boiled for 5 min at 100°C and separated on 12% SDS polyacrylamide gels.

## In-vivo UV-photocrosslinking analyses

The indicated *S.* Typhi strains harboring plasmids expressing the amber suppressor tRNAs and an orthogonal aminoacyl-tRNA synthetase were grown in TTIM for 24 hr at 37°C (to induce typhoid toxin expression and secretion), containing 10 μg/ml chloramphenicol and 1 mM of the unnatural photoreactive amino acid pBpa (p-benzoyl-L-phenylalanine, Bachem holding). Afterwards bacteria were split in two samples, one exposed to UV light and the other left untreated as control. For UV treatment, bacteria were transferred into sterile 60 mm petri dishes and placed under a UV lamp (365 nm) for 60 min. When applicable, samples were further incubated with lysozyme (200 μg/ml) for 30 min at 37°C in 10 mM Tris; pH eight buffer. Bacterial cells were collected by centrifugation, resuspended in low SDS (0.2%) Laemmli buffer, boiled, and analyzed by Western immunoblot blot with a mouse monoclonal antibody (Invitrogen) directed to the FLAG epitope present in TtsA.

## Crystallization and data collection

The purification of recombinant 6xHis-tagged Se-Methionine TtsA and 6xHis-tagged Sen1395 is described above. Initial spare matrix crystallization trials were carried out at the Yale University School of Medicine Structural Biology Core facility. After optimization, Se-Methionine TtsA crystals grew in one day at room temperature using the hanging-drop vapor diffusion method in a mix of 2 μl of protein (8 mg/ml) with 1 μl of reservoir solution consisting of 0.1 M sodium citrate pH 5.8, 0.1 M NaCl, and 18% v/v (+/-)−2-methyl-2,4-pentanediol. Native Sen1395 crystals grew in ~2 days at room temperature using the hanging-drop vapor diffusion method in a mix of 2 μl of protein (5 mg/ml) with 1 μl of reservoir solution consisting of 0.2 M sodium malonate pH 7.0% and 20% w/v PEG 3,350. In both cases the reservoir solution acted as cryoprotectant. Crystals were harvested after 5–10 days of growth and flash frozen in liquid nitrogen. Diffraction data for the Se-Methionine TtsA crystal were collected at Brookhaven National Laboratory (BNL) beamline AMX while data for the native Sen1395 crystals were collected at the Advance Photon Source beamline 24-ID. The data weas processed using HKL2000 (*Otwinowski and Minor, 1997*). The statistics are summarized in *Supplementary file 1* Table S1.

## Structure determination and refinement

The structure of TtsA was solved by single wave-length anomalous diffraction (SAD) phasing with data collected from a Se-Methionine crystal using Phaser-EP in Phenix (*McCoy et al., 2007*; *Adams et al., 2010*). The resulting electron density was of high quality and allowed for successful automated model building using Buccaneer (*Cowtan, 2006*). Iterative rounds of refinement in Refmac5 and Phenix were carried out (*Adams et al., 2010*; *Collaborative Computational Project, Number 4, 1994*; *Murshudov et al., 1997*) with manual rebuilding performed in COOT (*Emsley and Cowtan, 2004*). The structure of Sen1395 was solved by molecular replacement with Phaser in Phenix (*McCoy et al., 2007*; *Adams et al., 2010*) using the refined TtsA structure as the search model. As for TtsA, the model was refined using Refmac and Phenix (*Adams et al., 2010*; *Collaborative Computational Project, Number 4, 1994*; *Murshudov et al., 1997*) with manual rebuilding in COOT (*Emsley and Cowtan, 2004*). Figures for publication were prepared in Chimera (*Pettersen et al., 2004*).

## Acknowledgements

We thank the staff at the Advanced Photon Source beamlines 24ID-C and E and the National Synchrotron Light Source II beamline AMX. This research used resources of the National Synchrotron Light Source II, a U.S. Department of Energy (DOE) Office of Science User Facility operated for the DOE Office of Science by Brookhaven National Laboratory under Contract No. DE-SC0012704. The Life Science Biomedical Technology Research resource is primarily supported by the National Institute of Health, National Institute of General Medical Sciences (NIGMS) through a Biomedical Technology Research Resource P41 grant (P41GM111244), and by the DOE Office of Biological and Environmental Research (KP1605010). This research also used resources of the Advanced Photon Source, a U.S. Department of Energy (DOE) Office of Science User Facility operated for the DOE Office of Science by Argonne National Laboratory under Contract No. DE-AC02-06CH11357. This work was supported by National Institute of Allergy and Infectious Diseases grant AI079022 (to JEG).

## Additional information

### Funding

| Funder | Grant reference number | Author |
| --- | --- | --- |
| National Institute of Allergy and Infectious Diseases | AI079022 | Jorge E Galán |

The funders had no role in study design, data collection and interpretation, or the decision to submit the work for publication.

### Author contributions

Tobias Geiger, Conceptualization, Data curation, Formal analysis, Investigation, Visualization, Methodology, Involved in the execution and interpretation of all experiments shown except the solution of the atomic structures; Maria Lara-Tejero, Conceptualization, Data curation, Formal analysis, Investigation, Visualization, Methodology, Solved the atomic structures shown in this paper; Yong Xiong, Formal analysis; Jorge E Galán, Conceptualization, Resources, Formal analysis, Supervision, Funding acquisition, Project administration

### Author ORCIDs

Tobias Geiger (iD) https://orcid.org/0000-0002-8528-2849
Maria Lara-Tejero (iD) https://orcid.org/0000-0002-1339-0859
Jorge E Galán (iD) https://orcid.org/0000-0002-6531-0355

### Decision letter and Author response

Decision letter https://doi.org/10.7554/eLife.53473.sa1
Author response https://doi.org/10.7554/eLife.53473.sa2

## Additional files

### Supplementary files

- Supplementary file 1. Data collection and refinement statistics. Values in parenthesis are for the highest resolution shell.
- Supplementary file 2. Strains and plasmids.
- Transparent reporting form

### Data availability

The diffraction data have been deposited in PDB under accession code 6v40 and 6v3z. The rest of the data are included in the manuscript and associated supporting files.

The following datasets were generated:

| Author(s) | Year | Dataset title | Dataset URL | Database and Identifier |
|---|---|---|---|---|
| Lara-Tejero M, Galan JE | 2020 | Crystal structure of TtsA | https://www.rcsb.org/structure/6V40 | RCSB Protein Data Bank, 6V40 |
| Lara-Tejero M, Galan JE | 2020 | Crystal structure of Sen1395 | https://www.rcsb.org/structure/6V3Z | RCSB Protein Data Bank, 6V3Z |

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
