## [Decision Letter]

**Acceptance summary:**

This paper describes an exciting and rigorous set of experiments that help to elucidate how the typhoid toxin produced by *Salmonella* Typhi is secreted. The authors find that the muramidase TtsA is required for secretion and that TtsA is localized specifically to the cell poles by recognizing a specific form of the cell wall (peptidoglycan) found at the poles. The work not only sheds new light on how this particular toxin is secreted, but has implications for understanding how a variety of secreted products in bacteria are translocated across the bacterial cell envelope.

**Decision letter after peer review:**

Thank you for submitting your article "Mechanisms of substrate recognition by a typhoid toxin secretion-associated muramidase" for consideration by *eLife*. Your article has been reviewed by three peer reviewers, and the evaluation has been overseen by a Reviewing Editor and Gisela Storz as the Senior Editor. The following individual involved in review of your submission has agreed to reveal their identity: Kelly Hughes (Reviewer #1).

The reviewers have discussed the reviews with one another and the Reviewing Editor has drafted this decision to help you prepare a revised submission.

Summary:

This paper describes the structure and PG-binding and polar localization determinants of the toxin-secretion muramidase, TtsA. The authors identify a region of TtsA that is required for binding to L-D crosslinks in PG. This site is near the muramidase site, in the PG binding domain. They show that this site, N166, is required for binding to L-D crosslinked peptidoglycan. Their work shows that recognition of L-D crosslinked peptidoglycan and polar localization are mediated by different sites within the protein. They convincingly show that N166 is required for L-D crosslink recognition, and that the C-terminal domain more generally is required for polar localization. The work is rigorous, clever and nicely done, and the conclusions are exciting. This work sheds light on the function of TtsA, but also more broadly on how various protein domains interact with peptidoglycan. Most of the conclusions are supported by the data. However, there are a couple of places where the reviewers felt the wording should be adjusted to fit the data better. Additionally, the model proposed at the end was somewhat confusing and needs to be clarified.

Essential revisions:

1) Discussion, third paragraph – last few sentences. The conclusion that localization of TtsA to the poles is mediated by the activity of *ycbB* is not supported by the data, since TtsA still localizes to the poles in the ∆*ycbB* strain (Figure 2A). One possibility, that should be discussed, is that the L-D crosslinking is half of what localizes TtsA to the poles. It's not mentioned, but it seems that TtsA localization to the poles is reduced in the ∆*ycbB* strain.

2) The model shown in Figure 5E is confusing in light of the data in Figure 5C. From the right panel in C, it seems that N166, near the active site, is required for binding to the L-D crosslinks, but in the image in E, the L-D crosslinks are not near the N166 site. It is important to clarify the cartoon model more. N166 could be required for some conformational change in the protein that makes binding L-D crosslinks possible: there are ways that one can speculate that an amino acid is required for binding a region of the substrate that it doesn't touch it directly, but this isn't currently spelled out. The authors should clarify and better discuss this apparent contradiction in their model.

3) Related to the above statement – the role of the DAP binding site in the localization and function is not really explored. Perhaps tripeptides are also enriched at the poles? And the DAP binding helps localize TtsA to the poles? The data in Figure 5C very nicely show that N166 is involved in recognizing L-D crosslinks, but from the discussion it sounds like the authors are proposing that the rest of the C-terminal domain, including the DAP binding site, are also involved in recognizing L-D crosslinks. It's not clear the data back this up. It would be really nice to see how the D118TAG mutant binds to PG (i.e., assay of Figure 5C, right) in the ∆*ycbB* strain. If it still binds to PG in the ∆*ycbB*, then I think a more likely model would be that TtsA simultaneously binds to both L-D crosslinks at the N166 region, and to DAP residues in free tripeptides in the D118/ I116 region. However, perhaps this isn't the model, in which case, please make it more clear, and explain why N166 is not near the L-D crosslink site in the picture.

4) Throughout the manuscript, it is not entirely clear which experiments mainly confirm findings of two earlier manuscripts – Hodak et al., 2013 and Geiger et al., 2018 – and which ones represent new findings. For example are the authors replacing the earlier vesicle formation experiments with largely equivalent toxin secretion assays? This applies both to the actual results and their discussion, in particular with respect to Hodak 2103, which already showed the crucial role of the C-terminus of TtsA/Sen1395 and particularly amino acids 166/167 in function.

5) There are numerous inconsistencies and errors in the statistical analysis of the data.

6) In some cases, it is not obvious whether results were obtained for chromosomally encoded mutants, complemented deletion mutants, or mutant proteins co-expressed with the wild-type protein (some of these cases are listed below).

---

## [Author Response]

Essential revisions:1) Discussion, third paragraph – last few sentences. The conclusion that localization of TtsA to the poles is mediated by the activity of ycbB is not supported by the data, since TtsA still localizes to the poles in the ∆ycbB strain (Figure 2A). One possibility, that should be discussed, is that the L-D crosslinking is half of what localizes TtsA to the poles. It's not mentioned, but it seems that TtsA localization to the poles is reduced in the ∆ycbB strain.

We regret that the wording of the sentence in the Discussion may have caused some confusion. Indeed, we concluded that TtsA localization to the poles is NOT, at least exclusively, mediated by the activity of YcbB precisely because we observed polar localization of TtsA in the *∆ycbB* mutant, although as the reviewer points out, at a reduced level. However, in the sentence in question we did suggest that the activity of YcbB may result in the presence of PG cross-linked trimers and tetramers at the poles, which in turn may contribute to restrict TtsA’s enzymatic activity at this location since our crosslinking data suggest that TtsA must require complex PG crosslinking for its recognition. We do not believe that this statement is contradictory to the observation that TtsA localization does not require YcbB because protein localization and the restriction of the enzymatic activity may well be mediated by different mechanisms as our data suggest. We have edited the sentences in the Discussion to clarify this point.

2) The model shown in Figure 5E is confusing in light of the data in Figure 5C. From the right panel in C, it seems that N166, near the active site, is required for binding to the L-D crosslinks, but in the image in E, the L-D crosslinks are not near the N166 site. It is important to clarify the cartoon model more. N166 could be required for some conformational change in the protein that makes binding L-D crosslinks possible: there are ways that one can speculate that an amino acid is required for binding a region of the substrate that it doesn't touch it directly, but this isn't currently spelled out. The authors should clarify and better discuss this apparent contradiction in their model.

In the cartoon model we did not intend to imply any precise localization of any residue of TtsA, which is of course better depicted in the structure itself (we could have just as well depicted TtsA as any object). Rather, our intention was to capture in the cartoon the notion that TtsA requires a complex PG crosslinking pattern to recognize its substrates as suggested by our crosslinking experiments. Indeed, we observed that after in vivo and in vitro crosslinking, a large proportion of TtsA migrates as a dimer in SDS-PAGE, although TtsA cross-linked in the absence of PG migrates as a monomer. The chemistry of the chemical crosslinking reaction, both in vitro and in vivo, predicts that TtsA must be crosslinked to the free primary amino group of the two amino groups available in mDAP. Consequently, the appearance of a dimer in the crosslinking experiments indicates that the TtsA monomers must be crosslinked to adjacent peptide stems crosslinked to one another through complex crosslinking patterns such as trimers and tetramers since lysozyme-treated PG with canonical LD-crosslinks should yield unlinked TtsA monomers. As stated, in the cartoon our objective is to capture this complex issue but not to depict the location of any residue of TtsA, which is of course better depicted in the structure itself. We have clarified this issue in the figure legend.

*3) Related to the above statement* – *the role of the DAP binding site in the localization and function is not really explored. Perhaps tripeptides are also enriched at the poles? And the DAP binding helps localize TtsA to the poles? The data in Figure 5C very nicely show that N166 is involved in recognizing L-D crosslinks, but from the discussion it sounds like the authors are proposing that the rest of the C-terminal domain, including the DAP binding site, are also involved in recognizing L-D crosslinks. It's not clear the data back this up. It would be really nice to see how the D118TAG mutant binds to PG (i.e., assay of Figure 5C, right) in the ∆ycbB strain. If it still binds to PG in the ∆ycbB, then I think a more likely model would be that TtsA simultaneously binds to both L-D crosslinks at the N166 region, and to DAP residues in free tripeptides in the D118/ I116 region. However, perhaps this isn't the model, in which case, please make it more clear, and explain why N166 is not near the L-D crosslink site in the picture.*

Again, I think the reviewer is reading too much in our cartoon model, which does not intend to provide any topological relation of any specific residue relative to the peptide stem but rather to highlight the notion that the PG stem recognized by TtsA must be the subject of complex crosslinking since at least a trimer would be required to obtain the crosslinking pattern we observed. How each one of the amino acids involve in the binding participate in the interaction with a complex PG structure is difficult to ascertain and unfortunately would not be settled with additional crosslinking experiments. As discussed above, we have attempted to clarify this issue in the figure legend and in the text.

4) Throughout the manuscript, it is not entirely clear which experiments mainly confirm findings of two earlier manuscripts – Hodak et al., 2013 and Geiger et al., 2018 – and which ones represent new findings. For example are the authors replacing the earlier vesicle formation experiments with largely equivalent toxin secretion assays? This applies both to the actual results and their discussion, in particular with respect to Hodak 2103, which already showed the crucial role of the C-terminus of TtsA/Sen1395 and particularly amino acids 166/167 in function.

While in Hodak et al. we suggested the importance of the C terminus of TtsA for its function, the limitations of the assay did not allow us to pursue these observations any further. We pointed this out in the original version of the manuscript and attempted to clarify it further in the revised manuscript.

5) There are numerous inconsistencies and errors in the statistical analysis of the data.

We have addressed the inconsistences in the revised version.

6) In some cases, it is not obvious whether results were obtained for chromosomally encoded mutants, complemented deletion mutants, or mutant proteins co-expressed with the wild-type protein (some of these cases are listed below).

With very few exceptions (e. g. the over-expression experiments depicted in Figure 1C), all the constructs were allele swaps. We have clarified the issue in the revised version of the manuscript.